



# Intercomparison of Cosmic-Ray Neutron Sensors and Water Balance Monitoring in an Urban Environment

Martin Schrön[1,C], Steffen Zacharias[1], Gary Womack[2], Markus Köhli[1,3,4], Darin Desilets[2], Sascha E. Oswald[5], Jan Bumberger[1], Hannes Mollenhauer[1], Simon Kögler[1], Paul Remmler[1], Mandy Kasner[1,6], Astrid Denk[1,7], and Peter Dietrich[1]

[1]Dep. Monitoring and Exploration Technologies, Helmholtz Centre for Environmental Research - UFZ Leipzig, Germany
[2]Hydroinnova LLC, Albuquerque, US
[3]Physikalisches Institut, Heidelberg University, Germany
[4]Physikalisches Institut, University of Bonn, Germany
[5]Institute of Earth and Environmental Science, University of Potsdam, Germany
[6]Institute of Geosciences and Geography, University of Halle-Wittenberg, Germany
[7]Dep. of Geosciences, University of Tübingen, Germany
[C]Correspondence: martin.schroen@ufz.de

**Abstract.** Sensor-to-sensor variability is a source of error common to all geoscientific instruments, which needs to be assessed before comparative and applied research can be performed with multiple sensors. Consistency among sensor systems is especially critical when the signal is an integral value that covers a large volume within complex, urban terrain. Cosmic-Ray Neutron Sensors (CRNS) are a recent technology that is used to monitor large-scale environmental water storages, for which

a rigorous comparison study of numerous co-located sensors has never been performed. In this work, nine stationary CRNS probes of type `CRS1000` were installed in relative proximity on a grass patch surrounded by complex urban terrain. While the dynamics of the neutron count rates were found to be similar, offsets of a few percent from the absolute average neutron count rates were found. Technical adjustments of the individual detection parameters brought all instruments into good agreement. Furthermore, the arrangement of multiple sensors allowed to find a critical integration time of 6 hours above which all sensors

showed consistent dynamics in the data and their RMSE fell below $1\%$ of gravimetric water content. The residual differences between the nine signals indicated local effects of the complex urban terrain at the scale of several meters. Mobile CRNS measurements and spatial neutron transport simulations in the surrounding area ($25\,\mathrm{ha}$) have revealed that CRNS detectors are sensitive to sub-footprint heterogeneity despite their large averaging volume. The paved and sealed areas in the footprint furthermore damp the dynamics of the CRNS soil moisture product. We developed strategies to correct for the sealed-area effect

based on theoretical insights about the spatial sensitivity of the sensor. This procedure not only led to reliable soil moisture estimation in drying periods, it further revealed a strong signal of interception and evaporation water that emerged over the sealed ground during and shortly after rain events. The presented arrangement offered a unique opportunity to demonstrate the CRNS performance in complex terrain, and the results indicate great potential for further applications in urban water sciences.



## 1 Introduction

The recent method of Cosmic-Ray Neutron Sensing (CRNS) combines the geoscientific research fields of cosmic-ray neutron detection and environmental hydrology (Desilets et al., 2010; Zreda et al., 2012). The instrument measures the natural radiation of epithermal neutrons in a few meters above ground which is highly sensitive to the abundance of hydrogen atoms in the surrounding area. As neutrons can penetrate the soil up to depths of $80\,\mathrm{cm}$, and are then able to travel several hundreds of meters in air, the unique feature of the CRNS technology is the large averaging volume (Köhli et al., 2015). As a consequence, complex structures like buildings, water pools, trees, and sealed areas are probably influencing the average signal. Thus, interpretation of the CRNS soil moisture product in complex terrain is a crucial part of the data analysis. Several previous studies challenged this task in a forested water catchment (Bogena et al., 2013), in an alpine environment (Schattan et al., 2017), and in mixed landscapes (Schrön et al., 2017). This work explores for the first time the effects of even more extreme heterogeneity in an urban environment on the soil moisture monitoring performance.

Sensor comparability studies are an important step towards joint usage of multiple sensors for scientific applications. In the field of cosmic-ray neutron sensing, multiple sensors have been used as calibrate the CRNS rover during mobile surveys (Dong et al., 2014), and to support regional hydrological or land-surface models at multiple places within a catchment (Rosolem et al., 2014; Baatz et al., 2016). For isolated studies with single sensors, differences in the absolute counting rate are of minor relevance, so long as the offset for a probe is constant through time. However, as soon as these instruments are applied in a joint manner or in a mobile mode, the normalization of their signal is a prerequisit to make consistent interpretations of individual sensor performances and uncertainties.

Intercomparison studies are a preferable way to find corrections for the sensor-specific biases and were successfully applied to point sensors of soil moisture (Walker et al., 2004; Kögler et al., 2013) or to remote sensing instruments (Su et al., 2013). Intercalibration is also one of the main challenges for the worldwide neutron monitor network (Bachelet et al., 1965; Moraal et al., 2001; Krüger et al., 2008). For example, observations of the same magnetospheric event measured with neutron monitors on three continents revealed clear discrepancies that were related to device-specific configurations (Chiba et al., 1975). Moreover, Oh et al. (2013) compared data from 15 neutron monitors in the same period and concluded that individual detrending correction factors were needed for coherent prediction performance. Regarding cosmic-ray neutron sensors, inherent detector sensitivity will always vary somewhat from unit to unit due to limitations in manufacturing, as well as neutron detector type and size utilized. Calibrations may be employed to normalize such differences from instrument to instrument, and also account for any residual instrumental configuration inconsistencies. For this reason, Baatz et al. (2015) have determined sensor-specific *efficiency* factors empirically for their collection of CRNS probes.

Observations of epithermal cosmic-ray neutrons, in a stationary or a mobile mode, have played an important role to understand hydrological processes, and to support agriculture and model development (e.g., Rivera Villarreyes et al., 2011; Zreda et al., 2012; Franz et al., 2015; Han et al., 2016; Peterson et al., 2016). The unique features of the sensor, however, could be also used to estimate representative water contents in urban areas. This is a challenging task for conventional, invasive sensors due to partly non-accessible areas, which are not only a problem in agricultural hydrology (Ruan et al., 2001). The currently in-





creasing urbanization results in great challenges for the functionality and sustainability based on ecosystem services for human well-being (Grimm et al., 2008; Dobbs et al., 2014). Urban areas influence the micro-climate, biodiversity and soil stabilization in a considerable manner. Therefore, the understanding of urban hydrological processes as well the water requirements of urban vegetation are particularly important. The proportion of sealed surfaces and the associated reduction of water infiltration
as well as the high evaporation of water ponded at sealed surfaces are also a major threat to urban environments and urban micro-climate (Starke et al., 2010). Fundamental requirements to assess und understand these processes are the quantification of the evapotranspiration processes from soil and vegetation. The most prominent measurement methods to quantify evapotranspiration and soil water balance are lysimeter, sap flow, eddy covariance, bowen ratio energy balance methods or remote sensing techniques in combination with modeling approaches (for details see Nouri et al., 2013). The disadvantages of these
methods are the unrepresentativeness of point measurements, or the low resolution and penetration depth of remote sensing techniques. Up to now, few methods are available to assess urban soil moisture (Wiesner et al., 2016), urban water evaporation (Narita, 2007), or ground water recharge in urban areas (Göbel et al., 2004). For this reason, the usage of CRNS methods to fill the gap between point measurements and large-scale measurements would be a promising approach, especially since cosmic-ray neutron sensors are non-invasive, autonomous, measure continuously, and require low maintenance. The capabilities of the
CRNS method to capture different components of the water cycle in air, soil, and vegetation (e.g., Baroni and Oswald, 2015), and to integrate over large areas, could probably make a major difference to classical urban water hydrology.

One of the objectives of this paper is to investigate the various components of sensor-to-sensor variability in a systematic way. Nine CRNS probes have been co-located in a small grass area from Feb to Aug 2014, and the effect of sensor permutation and of detector-operating parameters will be observed in three distinct periods (Phases I, II, III). This opportunity has been also
used to test the consistency of the sensor ensemble with regards to temporal aggregation. To further understand the influence of the complex terrain to the sensor performance, simulations and mobile measurements have been consulted to reveal spatial patterns at the meter-scale. We finally present a method to correct for these spatial influences, which is tested using independent soil moisture observations.

## 2 Material and Methods

### 2.1 Cosmic-Ray Neutron Sensors (CRNS)

Neutrons in the energy range of 10 to $10^4$ eV (epithermal/fast) are highly sensitive to hydrogen, which turns neutron detectors to highly efficient proxies for changes of environmental water content. Zreda et al. (2012) presented the established method of cosmic-ray neutron sensing as a non-invasive and promising tool for hydrology applications, while Köhli et al. (2015) provided details of the underlying physics, the lateral footprint of several tens of several hectares, and the sample depth of up to 80 cm.
Cosmic-ray neutron sensors of type `CRS1000` (Hydroinnova LLC, US) are commercially available in several configurations. The main components and configurations have been described by Zreda et al. (2012) and are labeled in Fig. 1a. Each system comprises one bare neutron detector sensitive to thermal neutrons and one moderated neutron detector sensitive to epithermal/fast neutrons, advanced neutron pulse detecting modules (NPM), and a robust datalogger with integrated telemetry.


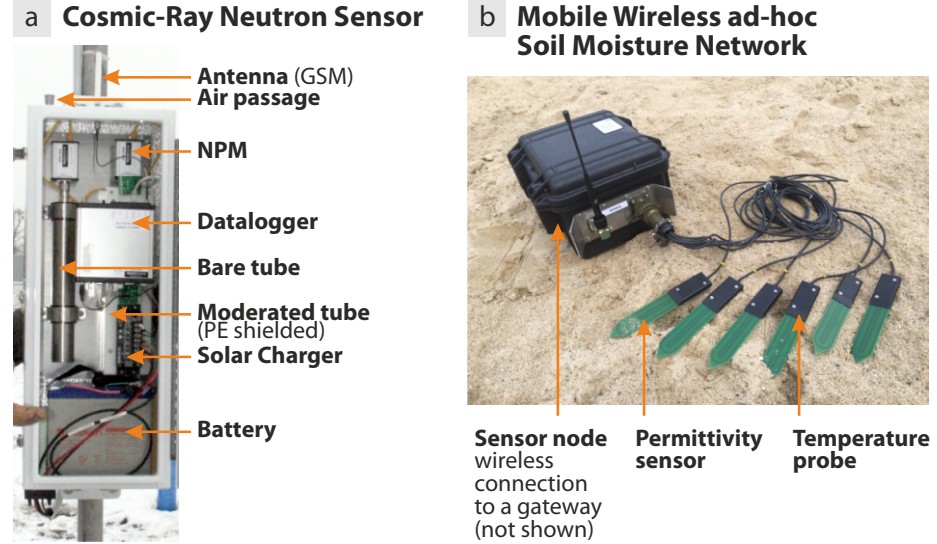

**Figure 1.** (a) Inside view of the cosmic-ray neutron sensor (CRNS) of type `CRS1000`. The *moderated* tube (surrounded by a white Polyethylene block) detects *epithermal* neutrons and is thus sensitive to water in the environment. (see the manufacturer's web page for additional information about the probe: http://hydroinnova.com/ps_soil.html#stationary). (b) The mobile ad-hoc sensor network node with six soil permittivity and soil temperature probes (`SMT100`, `TRUEBNER GmbH`, Neustadt, Germany), to be installed in different depths of the soil profile.

The logger retrieves neutron counts and diagnostic pulse height spectrum information periodically from each NPM, which generate the high voltage required by the detector tubes. The datalogger also samples barometric pressure sensors, and in this work, an external temperature and air humidity sensor (Campbell CS215, Campbell Scientific Inc., Logan, Utah, US). The datalogger has further been configured to record signals from additional external sensors, such as a tipping bucket rain gauge.
The mentioned devices and sensors are housed in a sealed metal enclosure.

The mobile variant of the CRNS detector is technically equivalent, but consists of larger gas tubes to increase the detection rate, which in turn allows for higher temporal resolutions (see also Chrisman and Zreda, 2013; Franz et al., 2015).

### 2.1.1 Neutron detection

A polyethylene shielded detector is employed to provide a moderated detector channel. The shielding material is designed to
10 reduce the number of incoming thermal neutrons and to slow incoming epithermal neutrons down to thermal (i.e., detectable) energies. An additional bare detector in the CRNS probe directly records incoming thermal neutrons, while it is less sensitive to epithermal energies.

Only thermal neutrons can be efficiently detected with state-of-the-art proportional detectors employing gases enriched in $^3$He (Persons and Aloise, 2011; Krane and Halliday, 1988). When a thermal neutron collides with an atomic nucleus of
15 the detector gas, a neutron absorption reaction can occur, resulting in emission of charged particles, which in turn produce





ionization. Electrons are attracted to the anode, a central wire at a potential of $\approx 1000\,\mathrm{V}$. Due to the steep gradient of the electrical potential towards the wire, the electrons are accelerated and collide with additional gas molecules, producing further ionization. A sensitive neutron pulse module (NPM), consisting of hybrid analog/digital electronics amplifies, shapes, and filters each charge pulse from the tube. The NPM further measures the pulse height, and records the pulse as a neutron count if

the pulse is a valid neutron event, and accumulates the pulse height in a pulse height spectrum.

### 2.1.2   Pulse Height Spectrum (PHS) recordings

As the energy of the reaction products in the gas is well known, a characteristic electronic pulse can be expected and translates to a prominent peak in a so-called *pulse height spectrum* (PHS), see for example Fig. 5b. However, sometimes the elements of the reaction product reach the wall of the tube before completely depositing their energy into the proportional gas. The

so-called *wall effect* is then visible in the PHS as a distribution of pulses of lower pulse height than the peak. As such, the typical shape of the pulse height spectrum is independent of the absorbed neutron energy. It is rather a function of the reaction kinematics and detector-specific details, including the geometry (Crane and Baker, 1991).

Pulse height spectra are autonomously and periodically recorded (typically daily or every several days) by the CRNS detector system, providing valuable self-diagnostics and longterm monitoring of the system health. An irregular PHS can have multiple

reasons, for example collapsing high-voltage supply (HV), gas leakage, or impurity in the detector tube, while variations at the lower end are an indication for current noise or gamma radiation. Typical CRNS systems have stable, longterm sensitivity to neutrons, and are maximally immune to environmental changes (such as temperature), electronic noise, and instrumental drift. More information about neutron detectors can be found for example in Mazed et al. (2012) and Persons and Aloise (2011).

### 2.1.3   The lower discriminator

The detector recognizes a neutron capture event if the released electronic pulse lies between the *lower discriminator* at the lower end and the *upper discriminator* at the upper end of the pulse height spectrum (beyond the prominent peak). The lower discriminator is an important detection parameter that is often set up on the "wall effect shelf" (the flat plateau in Fig. 5b), slightly to the right of the lower shelf edge (bins 30–35). This ensures maximum immunity to lower amplitude electronic noise which could otherwise be counted as neutron events. In addition, a high discriminator excludes signals from gamma

pileups which could otherwise produce spurious counts when in the presence of significant gamma radiation. However, the discriminator position above the shelf results in some loss of the theoretical maximum sensitivity of the neutron detector, and can cause some variation in sensitivity if the location of the lower discriminator relative to the peak location is not set consistently across multiple sensors.

Improvements in the NPM electronics since 2013 have increased the stability of the electronic gain and the high voltage

supply, as well as lowered the electronic noise floor. Therefore it is reasonable to use the whole pulse height spectrum for the neutron counter by setting the lower discriminator below the wall effect shelf (bins 23–24). One of the benefits is in maximally counting all neutrons (i.e., essentially counting very close to 100% of all neutron capture events). In addition, in



such a configuration, small changes in NPM electronic gain or internal high voltage will have the most minimal effect on the count rate.

## 2.2 Data processing and analysis

### 2.2.1 From neutrons to soil moisture

Nine cosmic-ray neutron sensors (CRNS) were deployed at a small grassland site (called *Schmetterlingswiese*) in the urban area of the UFZ, Leipzig, Germany (Figs. 2, 3). The instruments are typically equipped with sensors for air pressure $p_i$, air temperature $T_i$, and relative humidity $h_{\mathrm{rel},i}$. Their compound average, $\langle\cdot\rangle$, has been utilized to correct individual neutron count rates $N_{\mathrm{raw}}$ using standard procedures:

$$N_{\mathrm{corr},i} = N_{\mathrm{raw},i} \cdot \left(1 + \alpha\left(\langle h \rangle - h_{\mathrm{ref}}\right)\right) \cdot \exp\left(\beta\left(\langle p \rangle - p_{\mathrm{ref}}\right)\right) \cdot \left(1 + \gamma\left(I_{\mathrm{ref}}/I - 1\right)\right),$$

where $h(h_{\mathrm{rel}}, T)$ is the absolute humidity, $I$ is the incoming radiation (here: average signal from neutron monitors *Jungfraujoch* and *Kiel*), $h_{\mathrm{ref}} = 0\,\mathrm{g/m^3}$, $p_{\mathrm{ref}} = 1013.25\,\mathrm{mbar}$, $I_{\mathrm{ref}} = 150\,\mathrm{cps}$, and $\alpha = 0.0054$, $\beta = 0.0076$, $\gamma = 1$, (for details see Zreda et al., 2012; Rosolem et al., 2013; Hawdon et al., 2014; Schrön et al., 2015). The accepted approach to convert neutron count rates to (soil) water equivalent $\theta$ uses the following relation:

$$\theta(N) \cdot \varrho_{\mathrm{bulk}} = \frac{0.0808}{N/N_0 - 0.372} - 0.115 - \theta_{\mathrm{offset}}\,, \tag{1}$$

where $\varrho_{\mathrm{bulk}}$ (in $\mathrm{g/cm^3}$) is the soil bulk density, $\theta_{\mathrm{offset}}$ (in g/g) is the gravimetric water-equivalent of additional hydrogen pools (e.g. lattice water, soil organic carbon), and $N_0$ (in cph) is a free calibration parameter (for details see Desilets et al., 2010; Bogena et al., 2013).

According to Schrön et al. (2017) the CRNS probe does not measure a simple, equally-weighted average of the surrounding water content, due to the non-linearity of its radial sensitivity function, $W_r(h, \theta)$. Therefore, different parts of the footprint

area contribute differently to the average signal depending on their distance $r$ from the sensor. This knowledge can be used to quantify the contribution of individual areas that are of specific interest for CRNS calibration or validation purposes.

### 2.2.2 Counting statistics

As the grouped ensemble of nine sensors is a unique opportunity to frame the concept of a "super-detector", the combined signal will exhibit higher count rates and thus lower statistical noise. The average count rate $\langle N \rangle$ and its propagated uncertainty $\sigma$ of

each $i^{\mathrm{th}}$ sensor are given as:

$$\langle N \rangle = \frac{1}{9}\sum N_i\,, \quad \text{and} \quad \sigma(\langle N \rangle) = \frac{1}{9}\sqrt{\sum \sigma(N_i)^2},$$




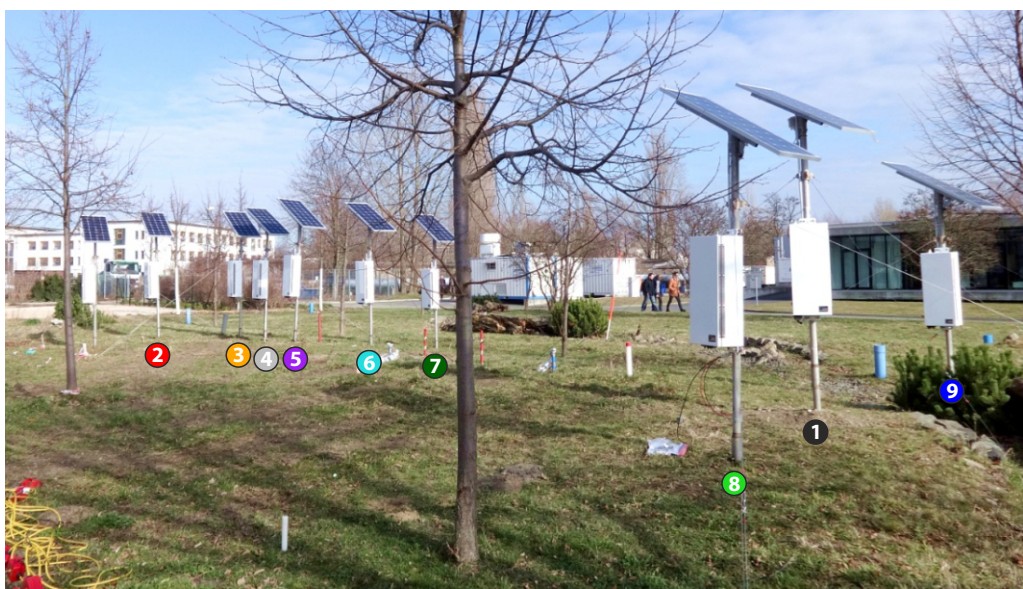

**Figure 2.** Arrangement of the nine stationary cosmic-ray neutron sensors in the small grass meadow surrounded by highly complex urban terrain.

where $\sigma(N) = \sqrt{N}$ is given as the standard deviation of average counts $N$ using Gaussian statistics. Under the assumption that $N_i \approx N_j \; \forall i, j \in (1, .., 9)$, their corresponding uncertainty will be similar as well:

$$\sigma(N_i) \approx \langle \sigma(N_i) \rangle \forall i, \quad \Rightarrow \quad \sigma(\langle N \rangle) \approx \frac{1}{9} \sqrt{9 \cdot \langle \sigma(N_i) \rangle^2} = \frac{1}{3} \langle \sigma(N_i) \rangle \approx \frac{1}{3} \sqrt{\langle N \rangle}$$

Hence, the combination of nine sensors reduces the relative statistical error by $67\%$, thereby allowing for accurate measure-
5  ments of changes of the environmental water storage. Temporal aggregation can further reduce the standard deviation. When the measurement interval $\tau$ (typically in counts per hour, cph) is aggregated to a longer period, $\tau_a = a\,\tau$, and then transformed back to units of $\tau$, the average count rate and uncertainty becomes:

$$N_a = \frac{1}{a} \sum_1^a N \approx \langle N \rangle, \quad \text{and} \quad \sigma(N_a) = \frac{1}{a} \sqrt{\sum_1^a \sigma(N)^2} \approx \frac{1}{\sqrt{a}} \langle \sigma(N) \rangle.$$

As a consequence, the average statistical error of a *daily* aggregated time series (in units of cph) is given as $\sigma(\langle N \rangle) =$
10  $\sqrt{\langle N \rangle / 24}$, which corresponds to 80% less uncertainty compared to hourly resolution.



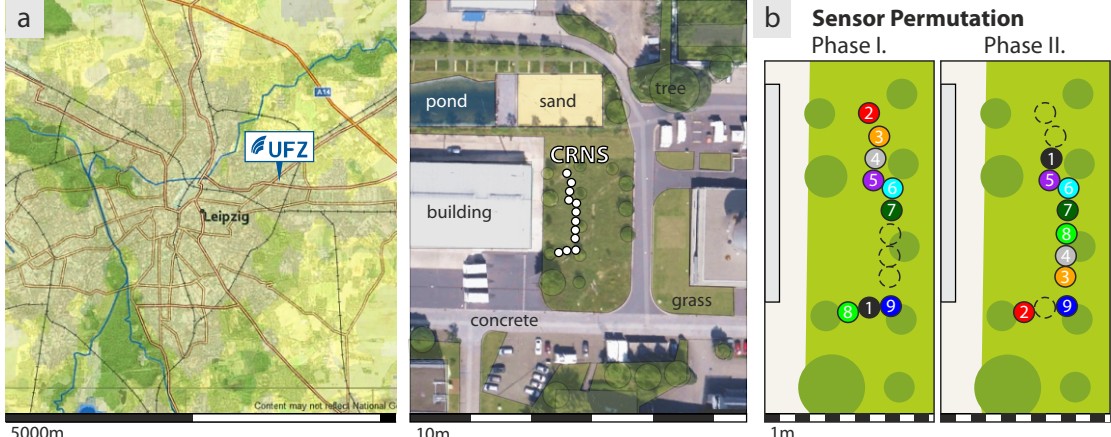

**Figure 3.** Location of the nine CRNS detectors deployed at the small meadow in the UFZ area. In March 2014, positions of some sensors were switched to test a hypothetic positional effect.

### 2.2.3 Performance measures

Generally, deviation measures can be expressed as an average of the individual vector $p$-norms (not to confuse with air pressure $p$) of all nine sensors:

$$\sigma^p(N) = \left( \frac{1}{9} \sum_i |N_i - \langle N \rangle|^p \right)^{\frac{1}{p}}, \quad \text{where} \quad \langle N \rangle = \frac{1}{9} \sum_i N_i \,.$$

5   The *standard deviation*, $\sigma^{p=2}(N)$, is used in Fig. 4 to measure the spread of individual sensors around their average $\langle N \rangle$. If not explicitly indicated, let $\sigma \equiv \sigma^{p=2}$ in this work. For two time series $N_1(t)$ and $N_2(t)$ with standard deviations $\sigma_1$ and $\sigma_2$, the *Pearson correlation coefficient* is defined as:

$$\rho(N_1, N_2) = \frac{\text{Cov}(N_1, N_2)}{\sigma_1 \sigma_2} = \frac{\langle (N_1 - \langle N_1 \rangle) \cdot (N_2 - \langle N_2 \rangle) \rangle}{\sigma_1 \sigma_2}$$

For example, $\rho = 0.7$ depicts that $N_1$ and $N_2$ can explain $0.7^2 \approx 50\,\%$ of their respective variance. If those two variables, 10   $N_1$ and $N_2$, were ranked depending on the order of their magnitude, $N_i \mapsto \text{Rank}(N_i)$, the *Pearson correlation* turns to the so-called *Spearman rank correlation*:

$$\rho_S(N_1, N_2) = 1 - 6 \frac{\sum_t \left( \text{Rank}(N_1) - \text{Rank}(N_2) \right)^2}{n(n^2 - 1)},$$

where $n$ is the number of days and $t \in (1, .., n)$. This quantity can be used to identify events that changed the rank of specific sensors.





## 2.3 The neutron transport simulator `URANOS`

The generation, interaction, and detection of neutrons can be simulated with Monte-Carlo codes, which are based on physically modeled interaction processes, and state-of-the-art cross section databases. The Ultra Rapid Adaptable Neutron-Only Simulation (`URANOS`) has been specifically tailored to environmental neutrons relevant for CRNS research. The model was described

by Köhli et al. (2015) to calculate the footprint volume and spatial sensitivity of CRNS probes. It has since been successfully applied to advance the method of cosmic-ray neutron sensing (Schrön et al., 2015; Schrön et al., 2017) and also to advance research in detector characterization for nuclear physics (Köhli et al., 2016). One of the unique features is the simulation of spatial neutron densities in an arbitrary, user-defined terrain. `URANOS` is very flexible and allows to input spatial bitmap information about the materials and geometries in the studied area. The software comes with a graphical user interface and is freely

available (see www.ufz.de/uranos).

## 2.4 The Mobile Wireless ad-hoc Soil Moisture Network (WSN)

In order to validate and calibrate the sensors against real soil water content, two independent measurement methods were consulted to quantify soil moisture profiles: volumetric soil samples (single measurement), and a *Mobile Wireless Soil Moisture Network* (continuous). The latter instrument is shown in Fig. 1b. The measurements were taken in different depths at two

locations near the CRNS probes #7 (north) and #9 (south). The corresponding soil parameters and soil moisture values (Table 1), and time series (Fig. 9) have been utilized to calibrate the neutron signal on volumetric soil moisture using eq. 1. The use of Wireless Sensor Networks (WSN) for monitoring applications are a promising tool in the field of environmental science to detect and record energy and matter fluxes across Earth's compartments (Hart and Martinwz, 2006; Zerger et al., 2010; Corke et al., 2010). The WSN used in this study has been developed specifically for short-termed, demand-driven applications

(Mollenhauer et al., 2015; Bumberger et al., 2015).

The soil moisture sensors of type `Truebner SMT100` used in the soil profiles directly measure electrical permittivity, $\varepsilon$, which is a compound quantity of the individual media (water, soil, air) and their volumetric fractions in the soil (Brovelli and Cassiani, 2008). The volumetric water content $\theta$ has been deduced from $\varepsilon$ with the *CRIM formula* (Roth et al., 1990), using independent measurements of porosity and soil water temperature, and assuming randomly aligned microscopic soil

structures. The official measurement uncertainty of ring oscillators is $1$–$2\,\%_{\mathrm{vol}}$, it can vary from wet to dry conditions, and is highly dependent on proper calibration against water and soil. Besides the device-specific uncertainty ($< 2\,\%_{\mathrm{vol}}$), the soil moisture profile measurements were also prone to inappropriate assumptions on the permittivity of quartz ($< 3\,\%_{\mathrm{vol}}$), and to the heterogeneity of soil properties and composition in the meadow ($< 8\,\%_{\mathrm{vol}}$). The latter uncertainty has been tested by sampling soil moisture profiles at many places within the field, and is taken into account when WSN is compared to CRNS observations.





**Table 1.** Two soil profiles in the grass meadow sampled nearby the profiles of the wireless sensor network (WSN) on Jan 14[th], 2016. Samples were taken with core cutters of constant volume at three depths, oven-dried, and weighted according to standard procedures. The evaporated water content is given in units of volumetric percent ($\%_{\text{vol}}$).

| profile | depth | $\varrho_{\text{bulk}}$ in g/cm$^3$ | porosity $\Theta$ | moisture $\theta$ |
|---|---|---|---|---|
| South | 7–12 cm | 1.62 | 38 $\%_{\text{vol}}$ | 18 $\%_{\text{vol}}$ |
| South | 15–20 cm | 1.52 | 42 $\%_{\text{vol}}$ | 15 $\%_{\text{vol}}$ |
| South | 25–30 cm | 1.58 | 40 $\%_{\text{vol}}$ | 18 $\%_{\text{vol}}$ |
| North | 5–10 cm | 1.60 | 40 $\%_{\text{vol}}$ | 32 $\%_{\text{vol}}$ |
| North | 15–20 cm | 1.93 | 27 $\%_{\text{vol}}$ | 19 $\%_{\text{vol}}$ |
| North | 25–30 cm | 1.91 | 28 $\%_{\text{vol}}$ | 28 $\%_{\text{vol}}$ |

## 3 Results and Discussion

### 3.1 Co-located sensor arrangement and permutation (Phases I+II)

In winter 2014, nine neutron detectors were co-located in the small grass meadow *Schmetterlingswiese* to probe the neutron density in air. Their individual count rates were logged every 15 minutes and processed using the standard correction approaches.

From Feb 22[nd] to Mar 18[th], statistical significant offsets have been observed between the individual signals, particularly for sensors #3 and #4 (Fig. 4). The average deviation of all count rates from their ensemble mean exceeded the daily statistical error, $\sigma(N_{24h}) \approx \sqrt{600/24} = 5$, by a factor of two. As the maximum distance between the sensors was 15 m, it has been hypthesized that the individual locations could have introduced a systematic effect on the count rate.

In the second phase, running from Apr 7[th] to May 8[th], positions of a subset of sensors were swapped, while others remained fixed (see Fig. 3b). In order to assess the effect on their individual measurement offsets, *Spearman rank correlations* have been applied to the time series before and after sensor permutation (see Fig. 5a). This quantity explains the probability with which a sensor's count rate $N$ is assigned to an ordered rank among the ensemble. The data showed that the favored rank (or offset) of both, fixed and swapped sensors, were almost unaffected. In particular, the ranks of sensors #3 and #4 remained at their high or low levels, respectively. Furthermore, the deviations of the sensor signals from their ensemble mean did not change significantly between phase I and II as plotted in Fig. 4. Following these argumentations, it can be concluded that small-scale positioning has not been the root course of the individual variability.

### 3.2 Adjustment of detection parameters (Phase III)

The previous experiment indicated that the observed sensor-to-sensor variability must have originated from device-specific differences. Therefore, Phase III has been dedicated to the diagnosis of the count rate, which is directly related to the integral of the *pulse height spectrum* (PHS). This supplementary data is automatically recorded during sensor operation and can be





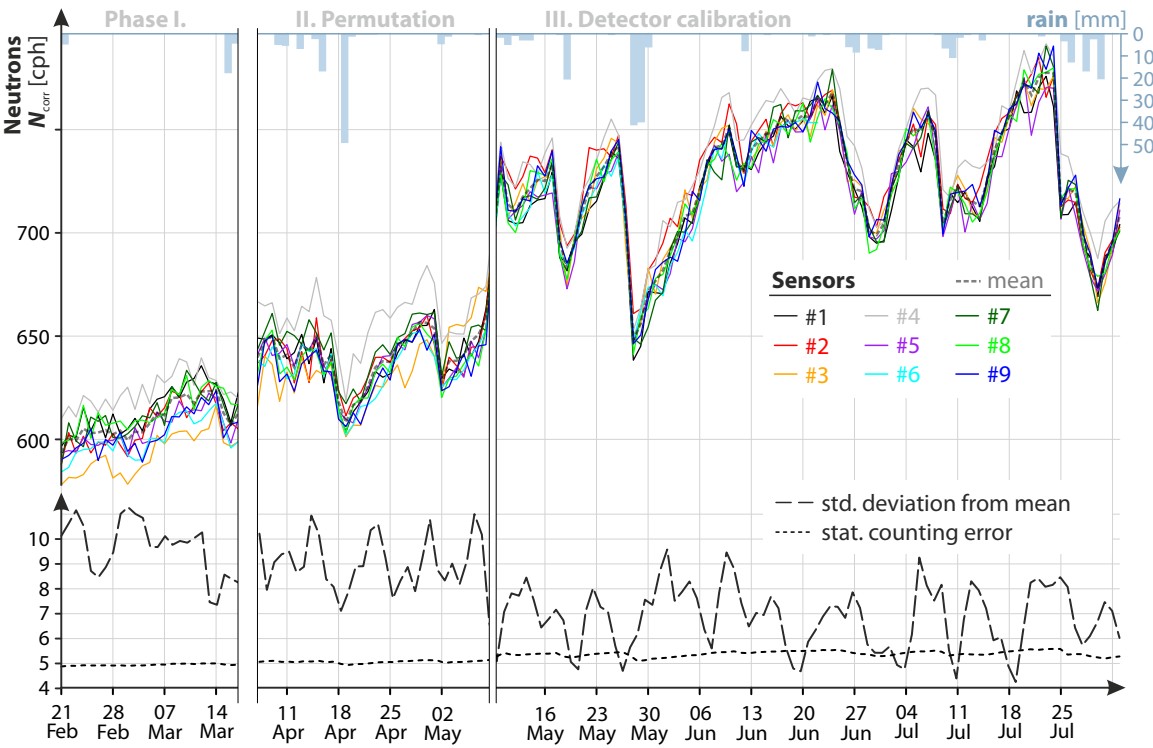

**Figure 4.** Time series of nine sensors covering phases I (installation), II (permutation) and III (calibration) in year 2014. By removing detector-specific effects in phase III, the standard deviation of the sensor ensemble from their mean could be reduced down to the statistical error of $\sigma \approx 5\,\mathrm{cph}$.

helpful for identification and interpretation of irregularities in the detection signal. As explained in section 2.1.2, the shape of the PHS and the parameters used to determine its integral (such as the *lower discriminator*) are important for the individual sensor efficiency. Thus, consistent detection parameters are a prerequisite to assure that the same fraction of neutron capture events are counted by all detectors. Fig. 5b indicates that this requirement was not met before Phase III.

5      During final factory configuration of the CRNS probes utilized in this work (i.e., brands prior to 2014), the manufacturer of the CRNS probes ensured maximum exclusion of electronic noise and gamma rays by setting the lower discriminator a bit up the wall effect shelf (around bin 30 to 35). However, its position in the spectrum (i.e., bin number) varied slightly from sensor to sensor, thus leading to small but measurable individual offsets in the count rate (Fig. 5b). Moreover, as year 2013 and later versions of the neutron pulse modules have an improved electronic noise floor and gamma pileup is not a concern in the probe

10     environment, lower discriminator values may be confidently used.

To achieve comparability of the sensors, we set the lower discriminator below the wall effect shelf (around bin 24) and adjusted the high voltage and amplifier gain parameters such that the main peaks aligned approximately to bin number 100 for the sake of visual accessibility. This procedure ensured maximum count rate for the individual sensors.



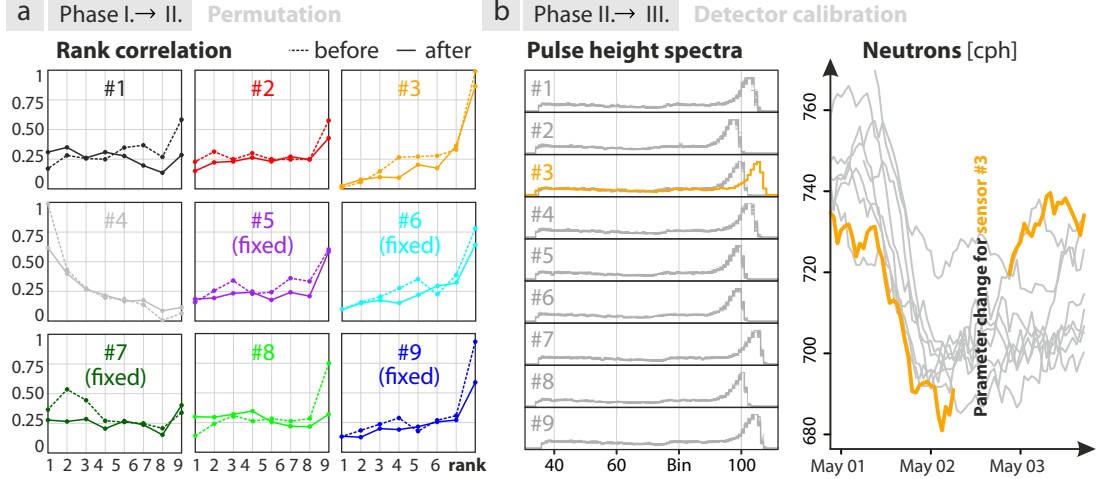

**Figure 5.** (a) Rank correlations of CRNS detector signals before and after permutation did not show significant change for swapped (non-fixed) sensors. (b) Calibration of the pulse height spectra and its impact on the count rate, exemplarily showing sensor #3 (orange).

In Fig. 5b the impact is demonstrated exemplarily for sensor #3. The parameter adjustments shifted the main PHS peak towards bin 100, and the reduction of the lower discriminator effectively increased the neutron count rate of the sensor. After manual adjustment of the parameters for all sensors, most of the individual offsets vanished and the standard deviation from the mean, $\sigma^{p=2}(N)$, has been reduced by $50\%$ down to the order of the statistical error (compare Fig. 4). Moreover, the average

absolute deviation, $\sigma^{p=1}(N)$, has been reduced even below the statistical error $\sigma(N) = \sqrt{N/24}$ of the daily aggregated time series (not shown). All in all, the instruments showed greater consistency in neutron counting sensitivity since the recovery of lower amplitude neutron pulse events that were previously being filtered by the lower discriminator.

### 3.3   Individual counting efficiency

The counting efficiencies (or relative variations around their mean) can be calculated by two methods. On the one hand, the

theoretical value can be determined by the relative positions of the lower discriminators in the PHSs. On the other hand, the empirical, more realistic value is inferred from the variability of the observed neutron counts. Fig. 6 shows the theoretical efficiencies of the nine sensors in Phase I, and their empirical values in Phases I, II, and III.

The results indicate that three distinct factors are contributing independently to the sensor-to-sensor variability. In Phase I, the theoretical (black) and empirical efficiency (red) deviate by $0.62\% \pm 0.18\%$. This indicates that the sensor-to-sensor

variability is a combined quantity that comprises more than the *detection-specific* variability from the PHS processor. The transition from Phase I (red) to Phase II (blue) introduces further changes of $0.45\% \pm 0.15\%$ for permuted sensors, but only $0.17\% \pm 0.06\%$ for fixed sensors. This trend indicates that there might be an additional component of the sensor-to-sensor variability that is related to location. Finally, the adjustment of the detector parameters at the beginning of Phase III causes



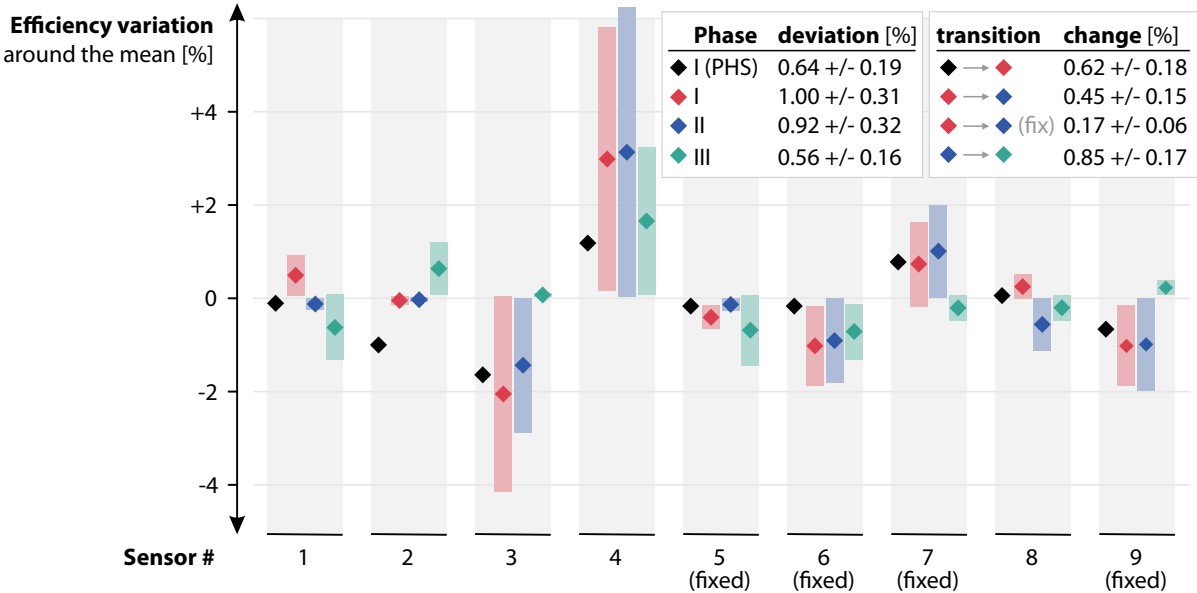

**Figure 6.** Relative variation of the neutron count rate around their ensemble mean, calculated before (Phase I) and after (Phase II) the swap of sensor positions, and after adjusting the detector parameters (Phase III). Additionally, the values for Phase I have been also determined just from the location of the discriminator in the pulse height spectrum (black).

the sensor efficiencies to change by $0.85\% \pm 0.17\%$. As the detector-specific variability was almost removed, the sensors now have the best agreement to each other.

The remaining variability could be contributed to small differences in design and geometry from the manufacturer, or the sensor location. The overall variability of $0.56\% \pm 0.16\%$ is now comparable with the standard relative error of the daily mean,

$\sigma(N)/N$, which went down to $0.5\%$ in certain periods of this study.

## 3.4   Temporal resolution for consistent observations

The previous sections have shown that the CRNS probes exhibited some small but measurable sensor-to-sensor variability that was related to the factory configuration of the neutron detector operating parameters. This section discusses the variability component that is related to statistical noise, which can be decreased by temporal aggregation. While Bogena et al. (2013)

were able to assess the appropriate temporal resolution of CRNS observations theoretically, the present arrangement provides a unique opportunity to test the approach with multiple sensors.

The ensemble-average *Pearson correlation* of the nine sensors significantly increased with increasing integration time across the three Phases. Fig. 7a shows that the correlation coefficient went up from $0.12$ and $0.26$ using 1 hour integration time, to $0.61$ and $0.74$ using 10 hours, for Phase I and Phase II, respectively. Since the sensor swap itself should have no effect to the

correlation, this effect can be attributed solely to the meteorological dynamics in these periods. While rain events were almost absent during Phase I (cmp. Fig. 4), the corresponding neutron dynamics were mainly influenced by statistical and detector-



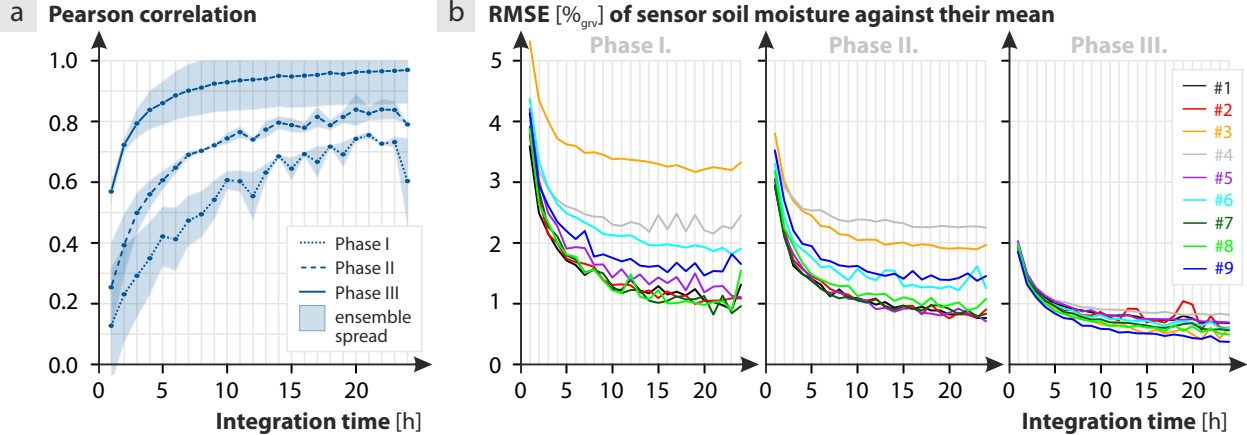

**Figure 7.** Influence of integration time to correlation and performance among the ensemble of nine sensors. (a) Ensemble-average Pearson correlation of the nine signals by twos for Phase I, II, III and temporal aggregation from 1 to 24 hours. (b) Root mean square error of the individual soil moisture products against the soil moisture product of the ensemble mean $\langle N \rangle$. Accuracy below gravimetric water contents of $1\,\%_{grv}$ can be achieved for all sensors when sensor-specific offsets were removed (Phase III) and the integration time exceeds 6 hours.

specific variability. In Phase II, a number of rain events lead to large amplitudes of neutron count dynamics and thus naturally to increased correlations.

Highest correlation has been achieved in Phase III, when most of the detector-specific variability has been removed (section 3.2). Moreover, correlation coefficients exceeded a value of 0.9 for more than 6 hours of integration time and went up to 0.97 for daily aggregation. These results demonstrate the reliability of CRNS observations for integration times of at least 6 hours under humid conditions, in complex terrain, and at sea level. Even higher correlations can be expected for dry regions and homogeneous terrain at high altitude, where higher neutron count rates would lead to lower statistical noise.

The accuracy of the CRNS soil moisture product also improves for higher integration times. In Fig. 7b the effect of the temporal aggregation of neutron counts is propagated to the individual soil moisture products $\theta(N_i)$, where their *root-mean-square-errors* (RMSE) against the ensemble mean $\theta(\langle N \rangle_i)$ is plotted. For all sensors, RMSEs can be reduced by 50–70 % using daily aggregation, while the accuracies of $1\,\%_{grv}$ gravimetric water content can be achieved beyond integration times of 6 hours. These findings agree quantitatively with theoretical calculations by Bogena et al. (2013), as well as similar experiments using Bonner spheres (Figs. 8–9 in Rühm et al., 2009).

### 3.5 Spatial heterogeneity in the footprint area

The previous sections have shown that positional effects in our study area can occur and should be taken into account, although their effect is less important than the detector-specific variability. Several of the conducted observations support the hypothetical influence of local effects within the complex terrain. For example, Fig. 4 shows high variability of neutron count rates in drying periods and low variability in wetting periods. This could be an effect of the dynamic size of the footprint. According to Köhli





et al. (2015), the distance which neutrons travelled before detection is smaller for wetter conditions. Thereby, distant structures could lose influence during and after rain events and thus would contribute to a harmonization of the nine sensor count rates. A second observation refers to Fig. 6, were noticeable changes of variability were observed for swapped sensors (Phase transition I→II), while the behaviour of fixed sensors were almost unchanged.

The two examples indicate that local effects might have the potential to influence the sensor performance. Local sensitivity of the neutron detectors has been augured already by Köhli et al. (2015) and could be a reasonable explanation given the heterogeneous distribution of the soil, of vegetation, and of nearby structures. This section tries to further quantify the local effects in a moisture-averaging footprint of several tens of hectares, where all sensors are exposed to similar meteorological forcings.

To assess the influence of complex terrain in the urban area, neutron transport simulations were conducted with the Monte-Carlo code URANOS (Köhli et al., 2015; Schrön et al., 2017). The urban scenario of $500 \times 500\,\mathrm{m}^2$ (Fig. 8a) has been re-enacted using 2D images of different layers that represent the different material compositions on the basis of their color code (illustrated in Fig. 8b as a compound image). Non-sealed area was defined as grassland with an exemplary soil moisture of $30\,\%_{\mathrm{vol}}$ and air humidity of $8.7\,\mathrm{g/m}^3$. The neutron density calculated by the simulation (Fig. 8c) clearly shows dry and wet features at the

meter scale that are related to the effects of buildings, sealed areas, the pond, iron-containing structures, and vegetation. Under these conditions it is evident that local heterogeneity in the footprint can have an effect on CRNS probes located within a few meters distance.

The URANOS model can help to assess those effects to support optimal sensor positioning or to explain unusual features in the spatial signal. The simulation results demonstrate the non-uniformity of the neutron density in the footprint. However, the

simulated quantities are not expected to exactly match reality due to many modeling assumptions that have been put into the scenario (clean material composition, uniform biomass density, homogeneous soil moisture). Nevertheless, the model results can be verified qualitatively with mobile CRNS measurements (Fig. 8d,e). Two surveys across parts of the urban area were conducted in May 2014 with a car and in July 2015 with a hand wagon. The neutron counts were collected along the track and logged together with GPS coordinates in one-minute intervals. A direct comparison with the simulation results was not

intended, as the low number of measurement points does not allow for meter-scale predictions of neutron density from the ordinary kriging interpolation. However, the collected data has been sufficient to support the theory of highly heterogeneous patterns in the urban terrain.

Both experimental and theoretical results clearly demonstrate that a significant neutron heterogeneity can occur within the CRNS footprint under conditions of complex terrain. These patterns have the potential to influence the CRNS measurements.

Moreover, slight variability is evident in the small meadow (center cross in Fig. 8), where trees and structures might influence the neutron density at the scale of a few meters. This could serve as an explanation for the minor position-related variability observed in the course of this study.





**Figure 8.** (a) Neutron environment of the urban CRNS test site (centered black cross). (b) Abstract model using geometric shapes, color-coded material definitions, air humidity $h = 8.7\,\mathrm{g/m^3}$, and soil moisture $\theta = 30\,\%_{\mathrm{vol}}$ in the grass areas. (c) URANOS simulation of neutrons detected in the $10\text{--}10^4\,\mathrm{eV}$ energy range. (d,e) Measured neutrons with the mobile CRNS Rover confirm heterogeneity of neutron patterns in the 0.1 ha installation area (0,0), as well as in the surrounding urban domain.





### 3.6 Areal correction for soil moisture estimation in partly sealed areas

Considering the revealed small-scale heterogeneity in the sensor footprint, as well as large sealed areas around the sensors, the important question arises whether CRNS in urban areas will be able to reliably estimate environmental water content. The reported footprint of the cosmic-ray neutron sensor covers approximately 6–17 hectares depending on wetness conditions (Köhli et al., 2015), which is a much larger area than the small meadow of $0.1\,\mathrm{ha}$ where the nine sensors are located (cmp. Fig. 9a). It can be expected that the paved and sealed areas beyond the meadow bias the integral soil moisture signal due to their inability to store water. This section tests the application of recent insights about the spatial sensitivity of CRNS probes, and demonstrates how this knowledge can help to understand – and even *correct* – the biasing effect of sealed areas. To support the theoretical calculations, the time series data of a wireless soil moisture monitoring network (WSN) has been compared with the CRNS soil moisture product in the small meadow. Furthermore, a water sprinkler is used to irrigate a small area near the CRNS probes and the effect on the neutron counts is examined.

Following theoretical considerations from Köhli et al. (2015) and robust evidence from Schrön et al. (2017), the radial sensitivity function $W_r(h,\theta)$ depicts the number of detected neutrons that originated from the distance $r$ under certain conditions of air humidity, $h$, and (soil) water equivalent, $\theta$. Its integral across all distances represents the total number of neutrons detected, $N$:

$$N = \int_0^\infty W_r(h,\theta) \cdot \mathrm{d}r.$$

A circular section of angle $\varphi$ (in radiant), which is confined between radii $r_1$ and $r_2$, contributes the following fraction of neutrons $n$:

$$n(r_1, r_2, \varphi) = \frac{1}{2\pi} \int_0^\varphi \int_{r_1}^{r_2} W_r(h,\theta) \cdot \mathrm{d}r \cdot \mathrm{d}\varphi' = \frac{\varphi}{2\pi} \int_{r_1}^{r_2} W_r(h,\theta) \cdot \mathrm{d}r. \tag{2}$$

The contribution area of the grassland meadow and surrounding patches is roughly equivalent to a circle of radius $r_2 \approx 20\,\mathrm{m}$. Hence, the portion of measured neutrons from this area is $n(0, r_2) \approx 41 \pm 2\,\%$, depending on $h$ and $\theta$. The dry and sealed areas beyond the grass meadow are effectively damping the dynamic signal from the meadow (see Fig. 9b).

To remove this damping effect, we suggest a new method to rescale the dynamic component of the neutron signal that is influenced by both, a variable and a constant patch in the footprint. At the urban test site, only $0.1\,\mathrm{ha}$ of the footprint contains soil beyond which everything else are either paved areas or solid buildings. Thus, only a small fraction $n(r_1, r_2, \varphi)$ of the total neutrons is connected to soil moisture variability. In order to compare these measurements with independent soil moisture sensors, we introduce an *areal correction*,

$$\text{areal correction:} \quad N' = C_{\text{area}}(N) = \frac{N - \langle N \rangle}{n(r_1, r_2, \varphi)} + \langle N \rangle, \tag{3}$$





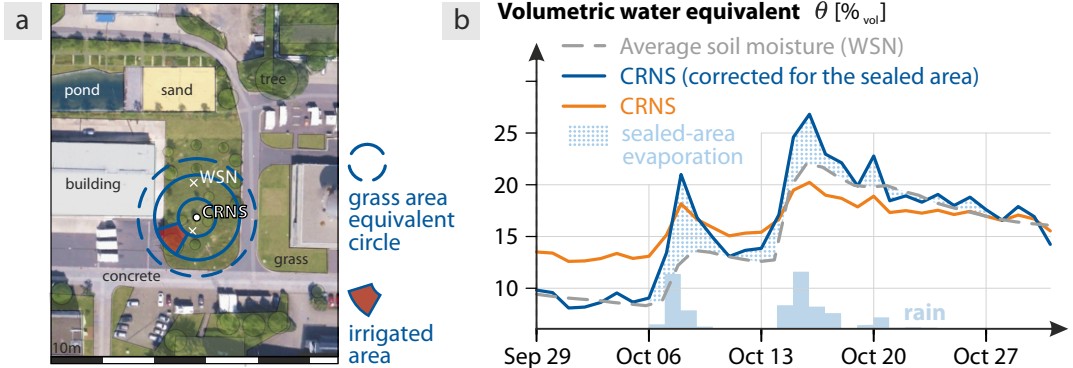

**Figure 9.** (a) Illustration of the circle (dashed blue) around the CRNS probe #7 that exhibits equivalent area to the non-sealed grassland meadow. The red shape covers the area irrigated by the sprinkler experiments. The white circle and crosses indicate the location of the CRNS probes and WSN profiles, respectively. (b) Demonstrating the area correction approach (eq. 3) in the small grass meadow. The constant contribution of the sealed area damps the soil moisture dynamics measured by the CRNS (orange line). The corrected signal (blue line) exhibits the rescaled dynamics based on the areal contribution of the meadow. Remaining deviation from the soil moisture profiles (grey dashed) during rain events represent intercepted water over the sealed ground.

that essentially scales the anomaly of neutrons by the inverse fraction of the contributing area. Using the data from CRNS probe #7 and the average soil moisture from a mobile soil moisture network, Fig. 9 demonstrates that this scaling approach brings both signals into good agreement and is therefore helpful to interpret CRNS data with confined areal coverage.

Besides the improved match of soil moisture dynamics, the area-corrected signal apparently overestimates soil moisture

peaks during rain events. This, however, is the representation of an important hydrological feature in urban areas. When the whole footprint is considered for data interpretation, it becomes evident that the precipitation water ponds on paved ground before it evaporates. Therefore, the additional water seen by the CRNS probe during rain events is the water that has been intercepted over sealed areas and that slowly evaporated in the subsequent hours.

The areal contribution theory can further help to estimate the sensitivity of the neutron sensors to dry or wet spots (see also

Schrön et al. (2017)). We conducted a sprinkler experiment in an area spanning an angle of $\approx 25°$ between 5 and 9 m distance from sensor #7 (red area in Fig. 9a). At two consecutive summer days, the ground was wetted once in the morning and once in the afternoon for a period of four hours each. With the help of eq. 2 the expected contribution of the sprinkled area can be estimated, which is $n(5\,\mathrm{m}, 9\,\mathrm{m}, 0.44\,\mathrm{rad}) \approx 0.26\,\%$ and as such far below the statistical significance. Thus, we did not expect the irrigation signal to be visible in the CRNS time series. Our data (not shown) confirmed this calculation, showing no significant

response of the neutron count rate to the small-scale irrigation periods.





## 4  Summary and Conclusion

The intercomparison study has provided a first impression of the systematic and statistical uncertainties related to neutron detection in the environment. Although the detectors were co-located within a small grass meadow at a total spatial extent of $15\,\mathrm{m}$ (while the footprint radius is at the order of $10^2\,\mathrm{m}$), their signals exhibited conspicuous differences in temporal variation and offset (Phase I). We investigated this behaviour experimentally by permuting some of the sensor locations (Phase II), and by re-adjusting sensor-specific detection parameters (Phase III). Thereby, individual efficiency factors have been determined that are needed to normalize the absolute count rates. To smooth out the remaining variability, we further tested various integration times of several hours to find an appropriate temporal resolution at which the nine sensors exhibit consistent behaviour. To understand the presence of positional effects caused by the complex terrain, neutron transport simulations and mobile neutron surveys were performed in the whole urban domain. We found evidence for sub-footprint heterogeneity of neutrons that contribute to bias and damping effects on the CRNS signal. Therefore, we developed a new correction method to allow for more accurate estimation of soil and intercepted water by taking into account the contribution of individual spatial structures to the average neutron signal. Five main insights have been obtained from these results:

1. Systematic variations in CRNS detector sensitivities in this work have been attributed to (1) detection parameters, and (2) small differences in detector manufacturing processes. The detection parameters have significant influence on the count rate and comparability of CRNS measurements (section 2.1.2). When detection parameters were adjusted with the assistance of the manufacturer (section 3.2), the total contribution of systematic errors in the intercomparison study was lowered to the order of the statistical counting error (section 3.3).

2. Some additional systematic error is present due to local positional effects. To assess the meter-scale heterogeneity of neutrons, the potential of `URANOS` spatial simulations has been demonstrated. A remarkable heterogeneity in the footprint has been revealed by simulations and confirmed by mobile CRNS measurements, which support the detector sensitivity to local effects.

3. If multiple standard CRNS detectors (`CRS1000`) are required to deliver similar results under similar conditions, a minimum temporal resolution of 6 hours was found to provide acceptable comparability for humid climate at sea level. The remaining statistical and systematic noise of the sensors could thereby be reduced below RMSEs of $1\,\%_{\mathrm{grv}}$ of gravimetric water content.

4. Soil moisture dynamics inferred from CRNS observations in partly sealed areas are significantly damped. An areal correction approach based on the sensitivity function $W_r$ (Köhli et al., 2015; Schrön et al., 2017) has been presented, that rescales the CRNS variability based on the contribution of the non-paved area in the footprint. This led to sufficient agreement with independently measured soil moisture profiles.

5. The CRNS probe is sensitive to intercepted and evaporated water over sealed areas. Such information can be used to actually quantify interception and evaporation processes (see, e.g., Baroni and Oswald (2015)), and could eventually contribute to closing the water balance in urban hydrology.



This work highlights the importance of studies on sensor-to-sensor intercomparison for geoscientific instruments. Those efforts can reveal unexpected features or systematic errors, can highly improve the understanding of the sensor response, and will thus improve their application in environmental sciences. One of the impacts of this study already led to improved efforts to adjust the detection parameters during the manufacturing process. The quantification of the sensitivity to local patches in

the footprint is particularly meaningful for supporting hyper-resolution land surface modeling (e.g., Chaney et al., 2016) and precision agriculture. The latter includes targeted irrigation based on critical information about soil properties, plant variety, and density (Hedley et al., 2013; Pan et al., 2013). In future studies we would recommend to further assess the potential of cosmic-ray neutron sensors for urban hydrology. Since water in complex terrain is almost impossible to quantify with point sensors, the large-scale averaging capabilities of cosmic-ray neutron probes can be a promising advantage for urban sciences.

*Acknowledgements.* U. Kappelmeyer (UFZ) provided meteorological data from a nearby UFZ owned weather station. We acknowledge the NMDB database www.nmdb.eu, founded under the European Union's FP7 programme (contract no. 213007) for providing data for incoming radiation, especially from monitors Jungfraujoch (Physikalisches Institut, University of Bern) and Kiel (Institute for Experimental and Applied Physics, University of Kiel). MS acknowledges kind support by the Helmholtz Impulse and Networking Fund through Helmholtz Interdisciplinary School for Environmental Research (HIGRADE). MK was funded by the Helmholtz Alliance EDA – Remote Sensing and

Earth System Dynamics, through the Initiative and Networking Fund of the Helmholtz Association, Germany. The research was funded and supported by Terrestrial Environmental Observatories (TERENO), which is a joint collaboration program involving several Helmholtz Research Centers in Germany.





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
