# Peer review of "Intercomparison of Cosmic-Ray Neutron Sensors and Water Balance Monitoring in an Urban Environment"

_Geoscientific Instrumentation, Methods and Data Systems, 2017_

## Referee Comment (RC1) · Anonymous Referee #2 · 11 Jul 2017

General comments:

The manuscript addresses relevant scientific questions suitable for GI. The manuscript covers two topics; 1) sensor inter-comparability, and 2) cosmic-ray neutron detection in an urban area. The study involves neutron transport modeling and field measurements using multiple stationary detectors as well as a cosmic-ray rover. In addition, soil moisture is estimated using frequency domain reflectometry sensors in various depths, and measured from soil sampling and oven drying.

Many aspects and applications are included in the manuscript, and it is obvious that a lot of work has gone into the making of the manuscript. Still, there is room for improve-

ment. As a reader I felt a little lost on what are the actual outcomes of the study. The authors touch upon a lot of different aspects, but the storyline is unclear. For example, I'm missing reasoning for: 1) why it is important to use multiple detectors for examining soil moisture in an urban area, 2) why it is important that detectors measure the exact same neutron intensity, and 3) why we would use CRNS detection to estimate area average soil moisture in an urban area with a lot of paved ground.

In order to avoid that the reader is left back with these unanswered questions, I suggest the authors to work on the storyline and the structure of the manuscript. At this point the introduction is a bit confusing and should be rearranged. Please introduce the manuscript by explaining why urban hydrology is important, what the problem is, and then in the end, how CRNS detection can be used to solve some of these problems. In the introduction, make sure to include an explanation to better link the part on inter-comparability and the part on soil moisture estimation in an urban area. In Section 2 the different equipment and modeling is described, however, some of the field experiments and modeling in also described in Section 3. I suggest the authors to collect all this in Section 2. Finally, I suggest that Section 3 is divided into two sections (one on "results", and another on "discussion").

The placement and numbering of figures should be changed. The figures should be placed at the point where they the first time are described. Furthermore, you should number the figures according to the order you are referring to them: Fig. 1, Fig. 2, Fig. 3. . ., and not like it is now: Fig. 1, Fig. 5, Fig. 4, Fig. 9. . . For these reasons I think the authors should put some more effort into improving the manuscript before publication.

Specific comments:

p1, l2-3: Please explain why consistency among sensors is more important when a larger volume is measured, and when a complex urban area is considered.

p1, l4: How large is "large-scale"? Please specify the CRNS response area.

p1, l9-10: Consider reformulating the sentence.

p2, l5: Because Köhli et al. (2015) report 83 cm, I suggest you to write "...of approximately 80 cm".

p2, l6: Desilets and Zreda, 2013 is an import paper on the examination of the neutron detector footprint, and must be included.

p2, l8-10: Include the studies of Franz et al., 2013 (WRR) and Franz et al., 2016.

p2, l9: "...in a forest (Bogena et al., 2015)".

p2, l10-11: This statement is out of place. Instead, the statement could be included it in the paragraph where you are listing the objectives of the paper (p. 3, l. 17-24).

p2, l12-15: Many more studies have used more neutron detectors at once. I suggest you to extent this part. This will also emphasize the importance of a sensor comparability study.

p2, l22: Explain "magnetospheric event".

p2, l30: Write "moderated cosmic-ray neutrons" instead of "epithermal cosmic-ray neutrons" as not only epithermal neutrons are detected.

p2, l33: But are we interested in the soil moisture underneath the buildings/roads? Please explain this.

p3, l1-2: The sentence should be rephrased, as it is hard to read.

p3, l1-6: The explanation of the problem of urbanization on hydrology should be improved. For example, why is urbanization challenging the functionality and sustainability, how is urbanization affecting the micro-climate, biodiversity and soil stability and why is a reduced infiltration and increased evaporation (because of sealed surfaces) a threat to urban environments and urban micro-climates?

p3, l6-7: Please explain why evapotranspiration (ET) is important. I would think that

infiltration is important because low infiltration at extreme rain events may result in flooding, which is harming the infrastructure and the buildings.

p3, l7-11: Eddy covariance measurements both represent a large area (similar to CRNS), and have a high temporal resolution. I would say that the disadvantage of Eddy is the price (equipment and maintenance). Furthermore, I don't understand the disadvantages of penetration depth in terms of ET measurements. Please revise the paragraph.

p3, l11-12: What is the normal practice in urban hydrology in terms of field work and modeling? Is it similar to what is done in rural areas? Are urban-hydrological models used? Please elaborate.

p3, l18-19: Too specific. Move this to "Methods".

p3, l12: Why is CRNS measurements useful in urban hydrology? Why do you want to know area-average soil moisture including the soil moisture below buildings and roads? Please explain.

p3, l18-19: For here, this sentence is too specific. Explain your objectives using more general terms, and consider moving the sentence to the section describing the method.

p3, l19-20: Please rephrase the sentence.

p3, l27-29: Not "while" as both Zreda et al. (2012) and Köhli et al. (2015) provided details of the underlying physics and the detector footprint. Zreda et al. (2008), Desilets and Zreda (2013) and Franz et al. (2012) are important papers and should be included.

p3, l32: You could consider shorten this sentence because you in Section 2.1.1 explain the energies measured by the detectors. For example: "Each system comprises a bare and a moderated neutron detector, an advanced NPM. . ."

p4, l4: Remove this sentence if you are not using this option in this study.

p4, l5: Please remove this sentence.

p4, l6-7: How is a larger tube increasing the counting rate? Please explain. Please describe the rover with greater detail, e.g., how many detectors (or the count rate compared to the stationary detectors), a map of the rout you are driving/walking, the speed you are collecting neutron intensity data and the dates of measurements. Some of this is described in Section 3.5. I suggest that the part from section 3.5 is moved up to the "method" section. The setup of the 9 stationary detectors is explained in Section 2.2.1. Please consider restructuring the section in a way so the setup and the field experiments conducted using the stationary detectors and the rover is explained concurrently.

p4, l12: Add a reference (Andreasen et al., 2016).

p5, l4-5: Revise the sentence - "puls" is mentioned five times.

p5, l8: Here the second reference to a figure is provided. Therefore it should be "Fig. 2b", not "Fig. 5b". Please fix the numbering of the figures throughout the whole manuscript.

p5, l12: Is this the geometry of the neutron detector? Please specify.

p5, l8-12: Is this a part of what sometimes make a PHS irregular (p5, l13-16)? If so, move this part down to there you are describing the sources for irregularity.

p5, l23: Please explain "bins". "Bins" of what?

p5, l24-25: Please provide a reference.

p5, l29: Are the NPMs of your systems from before or after 2013?

p5, l30-31: Can CRNS users themselves adjust the detector parameters, or is it something you need Quaesta/Hydroinnova to do for you? Later in the manuscript, I get the impression that you need the manufacture to adjust the detector parameters. However, please specify this a bit more clearly in the manuscript.

p6, l7: Consider to explain why neutron count rates are corrected for barometric pressure, atmospheric water vapor and incoming cosmic radiation. You should remove the abbreviations for air pressure (pi) and air temperature (Ti) as you are not using them later in the manuscript.

p6, l9: The equation is difficult to grasp. For the reader, it would be nice if you explained the different correction models one-by-one. In addition, numbering of this equation is missing.

p6, l11: What is "cps"?

p6, l17-20: This paragraph seems a little bit out of place. Either remove/move it, or better link it to the previous text, e.g., explain the calibration procedure (for the determination of N0).

p6, l22: Please explain what a super-detector is.

p6, l25 and p7, l1: Numbering of the equations is missing. This applies for most of the equations in the manuscript.

p7, l4 and l6: Is "a" the number of neutron detector? Please explain.

p7, l10: Consider to remove "(not to confuse with air pressure p)". Please extend the explanation of the performance measures using vector p-norms.

p8, Figure 3: You have a Figure 3a and a Figure 3b, and should therefore include "a" and "b" in the figure caption.

p8, l13: Please explain "cross-sections".

p9, l1: "(Schrön et al., 2015; 2017)". Note, not all journals accept a conference paper as a reference (Schrön et al., 2015).

p9, l2: What is meant by "advance research in detector characterization"?

p9, l6: The title should be changed as the section also describes volumetric soil sampling for soil moisture measurements.

p9, l9: What is meant by "continuous"? Are the sensors installed permanently at the two locations providing time-series of soil moisture? Or is soil moisture obtained on certain days of measurements? What time-period or what days did you measure?

p9, l9: What depths? Was the dry bulk density and the porosity calculated from the soil samples?

p9, l11-15: This paragraph could be taken out.

p9, l20-21: Please add a reference.

p9, l21-23: "device-specific" – is this for the Truebner SMT100 sensor? Please add a reference.

p9, l24: How many?

p9, l25: Please divide the section into two (one section on results, and another on discussion).

p9, l27-29: This should be described in the method-section.

p9, l28-29: Was the measured neutron intensity corrected for changes in atmospheric water vapor, pressure and incoming cosmic ray before performing the procedure obtaining the same count rate from all detectors? What is your reasoning for correcting before performing the comparability-procedure?

p9, l30: Should it say "In the first phase, from Feb 22nd..."? On page 11 line 1 you begin the sentence with "in the second phase,...", but at this point it is not clear what the first phase is. Furthermore, the different phases of the experiment should be described in the method section. In the results section, you merely provide the results of the experiment.

p11, l1-2: This should be moved to the method section.

p11, l12-15: Please remove this sentence as this is explained in section 2.1.2.

p11, l17-25: Please move this paragraph to the methods section.

p11, l23-24: How did you adjust these settings? Please explain this in the manuscript.

p11, l25: Could you quantify how much the procedure increased the count rate? I think many CRNS people will apply the detector parameter correction if the approach is shown to increase the neutron count rate significantly.

p12, Figure 5b: Please show the result of the calibration for all nine detectors (not only for sensor #3). In addition, the effect of correction could be quantified in counts and in percent (see comment above) and presented in a table.

p12, l2-5: This paragraph should be revised, and possible be moved to the section on methods.

p13, l11: What does cmp mean? Is that a standard abbreviation?

p14, l1: Consider to include a "field site" section in the beginning of the manuscript and include following details in the section: "humid conditions, in complex terrain, and at sea level".

p14, l2: Explain why a higher correlation is expected at homogeneous terrain. This should be included in the discussion-section.

p14, l13: Could this also be due to other reasons? E.g., the wetting occurs during short time periods (with similar weather conditions), while the drying occurs for multiple days (maybe) with changing weather, hence the variability is due to varying rates of evaporation and dewfall.

p14, l8: Most hydrologists will not know what a Bonners sphere is. Please include a description.

p15, l6-10: This belongs in the section describing the methods. In addition, what is the chemical composition of the material used for the paved ground (in terms of hydrogen content, is concrete "wet" or "dry" in comparison to the non-paved area of soil moisture

0.30)?

p15, l17: In order to give the reader a better insight to the modeling, please provide a detailed description of the model setup and the modeling procedure. On page 15 line 9-10 I got the impression that all non-sealed areas are defined as grassland, while it at page 15 line 16-17 sounds like vegetation is included (here "uniform biomass density" is mentioned as a modeling assumption). If vegetation was not included in the model, then the marks representing vegetation in Figure 8c should be taken out. If vegetation was included, then please provide some details on how vegetation was included in the model setup. E.g., amount of biomass (the hydrogen content) per tree, the chemical composition of the vegetation, vegetation height. . .

p15, l18: I suggest you to soften the sentence, as measurements and modeling were not directly compared, e.g., ". . .the model results can be assessed visually using measurements from a CRNS rover."

p15, l19-20: Move this to the "method" section. Furthermore, did you use the same equipment measuring from the car and the wagon, and were the detectors directed in the same way (horizontally or vertically) and installed in the same height? A map showing the CRNS roving route would be valuable (e.g., add the route in Figure 8). Please explain the field experiment in the "method" section.

p15, l21-22: The interpolation procedure for the measurements obtained using the CRNS rover should be explained in the "method" section.

p15, l20-23: Could the model results be scaled to match the CRNS roving measurements?

p15, l31: What is environmental soil moisture?

P16, Figure 8: What is the outline of the measured area in Figures 8D and 8E representing? Is it the driven route (I guess not, since the line crosses through building) or is it some meters away from where you drove/walked the rover (e.g., the footprint

radius)? Furthermore, what is the uncertainty of the measurements and modeling? Please provide these number in the figure or in the text.

p17, l1-2: Please revise the sentence, and remove the hyphens and the italic format.

p17, l4: This belongs to the "method" section.

p17, l26 – p18 l2: Please revise this paragraph. Interception, as an important variable for urban hydrology, should be mentioned when explaining urban hydrology in the introduction. The authors conclude that the difference between CRNS and WSN soil moisture is due to interception. Evidence is missing that the additional water seen by the CRNS detector is interception, and the difference could just as well be a result of other effects, e.g., different measurement scale (point scale and hectometer scale) and measurement depth. Either add some supporting modeling and measurement results or exclude the part of interception from the manuscript. A third option could be to include "interception" in the discussion of the results provided in Figure 9b. The discussion should also include other possible reasons for the different soil moisture of CRNS and WSN.

p18, l4-6: This belongs to the section describing the method.

p18, l3-9: Either, explain the purpose of the field experiment with greater detail as well as show and discuss the results, or take this part out.

p19, l4: See comment for "p17, l26 – p18 l2".

p19, l8: Please quantify (e.g., in %).

p19, l23-24: See comment for "p17, l26 – p18 l2".

p20, l1: If you use data retrieved from NMDB, NMDB ask you to: "acknowledge the origin by a sentence like "We acknowledge the NMDB database (www.nmdb.eu), founded under the European Union's FP7 programme (contract no. 213007) for providing data. ", and acknowledge individual monitors following the information given on the respective station information page."

p22, l10-19: The work by Köhli is not ordered alphabetically.

Technical corrections:

p2, l13: Should this be "...to calibrate the CRNS..."?

p2, l32: Should this be "..., could also be..."?

p3, l6: "and" instead of "und"

p3, l12: "groundwater"

p3, l33: Erase "robust".

p4, Figure 1 caption: Include the abbreviation PE for polyethylene in the figure caption. Remove the dot after "environment". Add primarily before "detects epithermal neutrons...".

p4, l2: Should it be "include a" instead of "samples"?

p5, l12: A dot is missing in the end of the sentence.

p6, l13: Should it be "soil moisture" instead of "(soil) water equivalent"?

p9, l7: I would say "calibrate and validate". Furthermore, I think you should remove "real" as soil moisture from the WSN is estimated from measures of permittivity. Only soil sampling and oven drying provides direct measures of soil moisture, and still this method has some uncertainties.

p10, Table 1: Should it be "soil moisture" instead of "evaporated water content"?

p11, l12: The abbreviation for PHS was already provided at page 5.

p14, Figure 7: "Influence of integration time, in hours (h),..."

p15, l7: "500m x 500m"

p17, l3: The abbreviation WSN was already introduced in section 2.4.

p17, l6: Erase "robust".

p17, l22: Remove "areal correction" (as it is also stated on line 21).

I hope you find the review useful.

———————————————

---

## Author Comment (AC1) · 25 Aug 2017

**Author Response to Reviews of**

**Intercomparison of Cosmic-Ray Neutron Sensors and Water Balance Monitoring in an Urban Environment**

M. Schrön, S. Zacharias, G. Womack, M. Köhli, D. Desilets, S. E. Oswald, J. Bumberger, H. Mollenhauer, S. Kögler, P. Remmler, M. Kasner, A. Denk, P. Dietrich

*gi-2017-34,* `doi:10.5194/gi-2017-34`
* * *
RC: *Reviewer Comment*,     AR: *Author Response*,     ☐ Manuscript text

*Dear Anonymuous Referee,*

*thank you for the positive and constructive review and the technical corrections to improve the manuscript. Please find our response to the main comments below:*

**1.  RC1 (Referee #2)**

**RC:** ***Many aspects and applications are included in the manuscript, and it is obvious that a lot of work has gone into the making of the manuscript. Still, there is room for improvement. As a reader I felt a little lost on what are the actual outcomes of the study. The authors touch upon a lot of different aspects, but the storyline is unclear. For example, I'm missing reasoning for: 1) why it is important to use multiple detectors for examining soil moisture in an urban area,***

 AR:  *In principle, water monitoring in an urban environment can be performed with a single cosmic-ray neutron sensor. In our study we tested the intercomparability and the effects of complex terrain to the sensors. As we aimed to resolve subtle features in the signal, we used a combination of multiple sensors to increase the measurement accuracy. Moreover, the complex environment is ideal to provide an upper-limit estimation of the sensor-to-sensor variability due to surrounding structures. We will elaborate on this in the revision.*

**RC:** *2) why it is important that detectors measure the exact same neutron intensity,*

 AR:  *Comparable efficiency of geoscientific instruments is a prerequisit for replicable research, especially when sensors are used in many different locations (see also more detailed explanations below). We addressed this question in the introduction section:*

> Sensor comparability studies are an important step towards joint usage of multiple sensors for scientific applications. [. . . ] For isolated studies with single sensors, differences in the absolute counting rate are of minor relevance, so long as the offset for a probe is constant through time. However, as soon as these instruments are applied in a joint manner or in a mobile mode, the normalization of their signal is a prerequisit to make consistent interpretations of individual sensor performances and uncertainties.

 *In particular, neutron detectors tend to measure different particles and energies when the detector parameters are not adjusted similarly among the probes. So the intercalibration performed in this study is not just based on an efficiency factor, it rather makes sure that the sensors actually measure the same quantity. We addressed this in the manuscript:*

> As explained in section 2.1.2, the shape of the PHS and the parameters used to determine its integral (such as the lower discriminator) are important for the individual sensor efficiency. Thus, consistent detection parameters are a prerequisite to assure that the same fraction of neutron capture events are counted by all detectors.

**RC:** *and 3) why we would use CRNS detection to estimate area average soil moisture in an urban area with a lot of paved ground.*

AR: *Water dynamics in a complex terrain (such as preferential flow, ponding, and intercepted water) are almost impossible to quantify with conventional point-scale sensors. The integral measurement signal of the CRNS probe may provide a better estimation of water storages under these circumstances. While we will improve the argumentation in the revision, some explanations were already given in the current manuscript:*

> Up to now, few methods are available to assess urban soil moisture (Wiesner et al., 2016), urban water evaporation (Narita, 2007), or ground water recharge in urban areas (Göbel et al., 2004). For this reason, the usage of CRNS methods to fill the gap between point measurements and large-scale measurements would be a promising approach, especially since cosmic-ray neutron sensors are non-invasive, autonomous, measure continuously, and require low maintenance. The capabilities of the CRNS method to capture different components of the water cycle in air, soil, and vegetation (e.g., Baroni and Oswald, 2015), and to integrate over large areas, could probably make a major difference to classical urban water hydrology.

**RC:** *In order to avoid that the reader is left back with these unanswered questions, I suggest the authors to work on the storyline and the structure of the manuscript. At this point the introduction is a bit confusing and should be rearranged. Please introduce the manuscript by explaining why urban hydrology is important, what the problem is, and then in the end, how CRNS detection can be used to solve some of these problems. In the introduction, make sure to include an explanation to better link the part on intercomparability and the part on soil moisture estimation in an urban area. In Section 2 the different equipment and modeling is described, however, some of the field experiments and modeling in also described in Section 3. I suggest the authors to collect all this in Section 2. Finally, I suggest that Section 3 is divided into two sections (one on "results", and another on "discussion").*

AR: *Thank you. We will clarify the argumentation in the introduction as suggested and move all descriptive parts from the results section to the methods section. We would like to keep Results and Discussion together as a single section, because the manuscripts deals with many subsequent experiments which are easier to follow in a coherent structure.*

**1.1. Page 1 L2–3**

**RC:** *Please explain why consistency among sensors is more important when a larger volume is measured, and when a complex urban area is considered.*

AR: *Thank you, we will rephrase the sentence to make it more clear. Consistency among sensor systems is important in every field of research, but especially when (1) multiple sensors are used to describe a single event, (2) multiple sensors are used in a joint application (e.g., data assimilation into models), or (3) subtle features in the signal are to be identified. Sensors with a large footprint volume in complex terrain are integrating many different signals (in this case, grass patches: areas of paved ground, trees, buildings, etc).*

*These factors can introduce subtle features in the sensor response, and can be different just due to the sensor position. In order to identify these features in the signal it is important to remove any instrumental noise or sensor-to-sensor variability.*

**1.2. Page 2 L12–15**

**RC:** *Many more studies have used more neutron detectors at once. I suggest you to extent this part. This will also emphasize the importance of a sensor comparability study.*

AR: *Thank you for this suggestions, we will expand this paragraph considerably by adding citations to other studies that require consistency among sensors, e.g. Andreasen et al. 2016, Evans et al. 2016, Hawdon et al. 2014, Franz et al. 2015.*

**1.3. Page 2 Line 22**

**RC:** *Explain "magnetospheric event"*

AR: *Magnetospheric events are short-term changes of the Earth's magnetic field as result of a solar plasma releases (coronal mass ejections), which eventually lead to changes of incoming cosmic radiation. Researchers found different response of different detectors on Earth although similar response was expected, which was partly related to detector-specific response functions. We will clarify this point and add citations.*

**1.4. Page 2 L33**

**RC:** *But are we interested in the soil moisture underneath the buildings/roads? Please explain this.*

AR: *Soil moisture beneath roads/buildings is not of interest in urban hydrology, and was not considered in our manuscript. Quantities of interest are water stored in soils (that are surrounded by paved areas), intercepted water in ponds or on leaves, and the corresponding potential and actual evaporation water. In complex terrain, areas are often not accessible, while distributed locations of intercepted water are hard to quantify. We will clarify this in the revision.*

**1.5. Page 3 L1–6**

**RC:** *The explanation of the problem of urbanization on hydrology should be improved. For example, why is urbanization challenging the functionality and sustainability, how is urbanization affecting the micro-climate, biodiversity and soil stability and why is a reduced infiltration and increased evaporation (because of sealed surfaces) a threat to urban environments and urban micro-climates?*

AR: *Thank you for emphasizing the need for further explanations. In the revision we will extend the describtion of the problems in urban hydrology by highlighting the findings from the cited studies.*

**1.6. Page 3 L6–7**

**RC:** *Please explain why evapotranspiration (ET) is important. I would think that infiltration is important because low infiltration at extreme rain events may result in flooding, which is harming the infrastructure and the buildings.*

AR: *Infiltration is indeed a factor for infrastructural problems, but also for groundwater recharge. Evaporation water influences air humidity, and thus the micro-climate, air quality, and temperature. Plant water transpiration is different in urban areas compared to non-urban areas due to different air temperature, winds, and*

*many more factors. It is important to understand these processes in order to quantify available water and to adapt planning of parks and buildings to allow optimal circulation of water and air. See also Arnfield, A. J. (2003). Two decades of urban climate research: a review of turbulence, exchanges of energy and water, and the urban heat island. International Journal of Climatology, 23(1), 1-26.). We will add more explanations about this aspect in the text.*

**1.7. Page 3 L7–11**

**RC:** ***Eddy covariance measurements both represent a large area (similar to CRNS), and have a high temporal resolution. I would say that the disadvantage of Eddy is the price (equipment and maintenance). Furthermore, I don't understand the disadvantages of penetration depth in terms of ET measurements. Please revise the paragraph.***

**AR:** *Thank you, the listing of disadvantages refered to soil water estimation, not Eddy-covariance. We will add the price of EC as a disadvantage and the whole paragraph will be clarified in the revision.*

**1.8. Page 3 L12**

**RC:** ***Why is CRNS measurements useful in urban hydrology? Why do you want to know area-average soil moisture including the soil moisture below buildings and roads? Please explain.***

**AR:** *Thank you, soil moisture estimation below roads/buildings is not of interest. Determination of available water in the urban environment is of interest instead, which includes water in near-surface soils (e.g., of grass patches), surface and intercepted water (e.g., in ponds or on leaves). This is a challenging task when water is intercepted by, or areas of interest are surrounded by, complex infrastructure like roads and buildings. We will clarify this aspect in the revision.*

**1.9. Page 4 L4**

**RC:** ***Remove this sentence if you are not using this option in this study.***

**AR:** *We used this feature to connect a tipping bucket to the datalogger. We thus like to stay with this sentence.*

> The datalogger has further been configured to record signals from additional external sensors, such as a tipping bucket rain gauge.

**1.10. Page 4 L6–7**

**RC:** ***How is a larger tube increasing the counting rate? Please explain.***

**AR:** *A neutron detector counts the number of interacting neutrons that entered the volume of the gas tube. The larger the detector, the higher the probability of a neutron to hit the detector gas. This increases the count rate, and thus reduces the measurement uncertainty of the neutron density in air. We will add explanations in the revision.*

**RC:** ***Please describe the rover with greater detail, e.g., how many detectors (or the count rate compared to the stationary detectors), a map of the route you are driving/walking, the speed you are collecting neutron intensity data and the dates of measurements.***

**AR:** *Thank you, we will add a few details about the experiment to the method section and add a line showing the*

*walked path to Figure 8. Since the rover is not the central part of this study, we think that short descriptions are sufficient while further references direct the reader to the details of the technique.*

RC: ***Some of this is described in Section 3.5. I suggest that the part from section 3.5 is moved up to the "method" section. The setup of the 9 stationary detectors is explained in Section 2.2.1. Please consider restructuring the section in a way so the setup and the field experiments conducted using the stationary detectors and the rover is explained concurrently.***

AR: *Thank you for the constructive suggestions, we will change the manuscript accordingly.*

**1.11.  Page 5 L8**

RC: ***Here the second reference to a figure is provided. Therefore it should be "Fig. 2b", not "Fig. 5b". Please fix the numbering of the figures throughout the whole manuscript.***

AR: *We have checked the figure references throughout the manuscript and find that they are correct. However, we admit that the illustration of a pulse height spectrum (PHS) as described here is probably worth an own figure, such that we will consider adding a new Figure 2. This will avoid references to Fig. 5 in this early part of the manuscript.*

**1.12.  Page 5 L8–12**

RC: ***Is this a part of what sometimes make a PHS irregular (p5, l13-16)? If so, move this part down to there you are describing the sources for irregularity.***

AR: *No, the wall effect is a typical and unavoidable feature of the pulse height spectrum. Its ideal shape is determined by constant factors like detector geometry or physics of the interactions between detector gas and neutron.*

**1.13.  Page 5 L23**

RC: ***Please explain "bins". "Bins" of what?***

AR: *Bins of released energy, we will clarify it in the text.*

**1.14.  Page 5 L30–31**

RC: ***Can CRNS users themselves adjust the detector parameters, or is it something you need Quaesta/Hydroinnova to do for you? Later in the manuscript, I get the impression that you need the manufacture to adjust the detector parameters. However, please specify this a bit more clearly in the manuscript.***

AR: *Theoretically, the user interface of the datalogger allows to change the parameters. However, according to the manufacturer, the parameters should be adjusted with guidance by Quaesta Instruments.*

**1.15.  Page 6 L7**

RC: ***Consider to explain why neutron count rates are corrected for barometric pressure, atmospheric water vapor and incoming cosmic radiation. You should remove the abbreviations for air pressure (pi) and air temperature (Ti) as you are not using them later in the manuscript.***

AR: *Thank you, we will remove the indices in this paragraph. We will also add an explanation that the measured*

*intensity of albedo neutrons depends on the intensity of incoming cosmic-ray neutrons, which changes with changing atmospheric conditions and changing intensity of incoming galactic cosmic rays.*

**1.16.    Page 6 L9**

**RC:**    ***The equation is difficult to grasp. For the reader, it would be nice if you explained the different correction models one-by-one. In addition, numbering of this equation is missing.***

AR:    *Thank you for this suggestion. The correction methods have been explained in high detail elsewhere, and since it is not the focus of this study, we think that it is sufficient to provide a brief summary of the applied corrections.*

**1.17.    Page 7 L4 and L6**

**RC:**    ***Is "a" the number of neutron detector? Please explain.***

AR:    *The variable $a$ denotes the factor by which a measurement interval is extended, i.e., the signal is aggregated. It is already indicated in the text, but we will add further clarification.*

> When the measurement interval $\tau$ (typically in counts per hour, cph) is aggregated to a longer period, $\tau_a = a\tau, \dots$

**1.18.    Page 7 L10**

**RC:**    ***Consider to remove "(not to confuse with air pressure p)". Please extend the explanation of the performance measures using vector p-norms.***

AR:    *Thank you, we will simplify the details about $p$ vector norms, as it boils down to basic statistical measure theory, which is not the focus of this paper. We are using the norms $p = 1$ and $p = 2$ in the manuscript to interpret the measurement error.*

**1.19.    Page 9 L3**

**RC:**    ***Please explain "cross-sections".***

AR:    *The term* cross-section *relects the interaction probability of an atomic nucleus with a neutron. It is the main control of the neutron intensity on Earth and thus of fundamental importance for neutron transport simulations. We will elaborate a little bit more on this topic.*

**1.20.    Page 9 L9 (L14?)**

**RC:**    ***What is meant by "continuous"? Are the sensors installed permanently at the two locations providing time-series of soil moisture? Or is soil moisture obtained on certain days of measurements? What time-period or what days did you measure?***

AR:    *The mobile soil moisture monitoring network was operated for several months and measured soil moisture continuously at time steps of several minutes. We will add details in the revision.*

**1.21.   Results and Discussion**

RC:   *Please divide the section into two (one section on results, and another on discussion).*

AR:   *We will consider this suggestion together with the opinion of the other reviewers. We think that such a split would reduce the readability of the manuscript, as the story is already linearly splitted into different subtopics that eventually lead to a final conclusion about the sensor efficiency and sensitivity. As an alternative, we will consider moving most of the experimental descriptions to the methods section, which would shorten and improve the readability of* Results and Discussions.

**1.22.   Page 9, L28-29**

RC:   *Was the measured neutron intensity corrected for changes in atmospheric water vapor, pressure and incoming cosmic ray before performing the procedure obtaining the same count rate from all detectors? What is your reasoning for correcting before performing the comparability-procedure?*

AR:   *Yes, we corrected the neutron counts as was described in section 2.2. of the manuscript. The three mentioned factors arise from meteorological changes that are unrelated to surface conditions and thus equal for all nine CRNS detectors. In principle, pure determination of efficiency would be possible without correction. However, the overall modulation of the signal with these external factors is of the same magnitude as soil moisture changes. Intercomparison measures, such as the correlation (Fig. 7), would lead to values close to 1 if the signals are modulated by the same factors to such a high degree. It would also be impossible to draw conclusions about the intercomparability of their soil moisture products.*

**1.23.   Page 9 L30**

RC:   *Should it say "In the first phase, from Feb 22nd..."? On page 11 line 1 you begin the sentence with "in the second phase,...", but at this point it is not clear what the first phase is. Furthermore, the different phases of the experiment should be described in the method section. In the results section, you merely provide the results of the experiment.*

AR:   *Thank you, we will describe the phases in more detail in the methods section. Our intention to also repeat some descriptions in the results section was to improve the readability of the manuscript.*

**1.24.   Page 11 L23-24**

RC:   *How did you adjust these settings? Please explain this in the manuscript.*

AR:   *Parameter adjustments are detector- and firmware-specific and do not contribute to the scientific message. We will add a note that consultation of the manufacturer is necessary to change the detector parameters.*

**1.25.   Page 11 L25**

RC:   *Could you quantify how much the procedure increased the count rate? I think many CRNS people will apply the detector parameter correction if the approach is shown to increase the neutron count rate significantly.*

AR:   *The procedure increased the count rate by 0–3 percent depending on the sensor. According to the manufacturer, these improvements have been already applied for newer detectors.*

**1.26.  Figure 5b**

RC:   *Please show the result of the calibration for all nine detectors (not only for sensor #3).*

AR:   *Thank you, we will consider adding the corrected pulse height spectra to the figure.*

*In addition, the effect of correction could be quantified in counts and in percent (see comment above) and presented in a table.*

*We presented the effect of the correction in Fig. 6, showing the relative variation around the mean before (blue) and after (green) parameter adjustment.*

**1.27.  Page 14 L1**

RC:   *Consider to include a "field site" section in the beginning of the manuscript and include following details in the section: "humid conditions, in complex terrain, and at sea level".*

AR:   *Thank you, we will add such information to the methods section.*

**1.28.  Page 14 L2**

RC:   *Explain why a higher correlation is expected at homogeneous terrain. This should be included in the discussion-section.*

AR:   *The study showed that complex terrain could introduce a positional effect to the sensors, as different wet and dry patches in the footprint as well as trees and buildings could disturb the signal. Therefore, higher correlation among sensors could be expected in homogeneous terrain where signals are undisturbed.*

**1.29.  Page p14 L13**

RC:   *Could this also be due to other reasons? E.g., the wetting occurs during short time periods (with similar weather conditions), while the drying occurs for multiple days (maybe) with changing weather, hence the variability is due to varying rates of evaporation and dewfall.*

AR:   *We agree to this explanation and thank you for the suggestion. It will be added to the text.*

**1.30.  Page 15 L6-10**

RC:   *This belongs in the section describing the methods. In addition, what is the chemical composition of the material used for the paved ground (in terms of hydrogen content, is concrete "wet" or "dry" in comparison to the non-paved area of soil moisture*

AR:   *The hydrogen water content of concrete corresponds to $\approx 10\,\%$ water content in our simulation, we will add details of the material composition in the methods section.*

**1.31.  Page 15 L17**

RC:   *In order to give the reader a better insight to the modeling, please provide a detailed description of the model setup and the modeling procedure. On page 15 line 9-10 I got the impression that all non-sealed areas are defined as grassland, while it at page 15 line 16-17 sounds like vegetation is included (here "uniform biomass density" is mentioned as a modeling assumption). If vegetation was not included in the model, then the marks representing vegetation in Figure 8c should be taken out. If vegetation was*

*included, then please provide some details on how vegetation was included in the model setup. E.g., amount of biomass (the hydrogen content) per tree, the chemical composition of the vegetation, vegetation height...*

AR: *Thank you for these suggestions. It is correctly written that non-sealed surfaces consist of grassland in the model. In addition to that, we also added vegetation, buildings, etc, above the surface, as shown in Fig. 8. While we will add a few more details of the simulation setup in the method section, we want to emphasize that the exact replication of the urban area was not intended in our study. The main focus was to show the spatial heterogeneity of neutrons in the footprint area, as a general contribution to the understanding of neutron distributions. Both, model and observations show such heterogeneity, while exact quantification is beyond the focus of this study. This was already written in the text:*

> The simulation results demonstrate the non-uniformity of the neutron density in the footprint. However, the simulated quantities are not expected to exactly match reality due to many modeling assumptions that have been put into the scenario (clean material composition, uniform biomass density, homogeneous soil moisture).

**1.32. Page 15 L19-20**

RC: *Move this to the "method" section. Furthermore, did you use the same equipment measuring from the car and the wagon, and were the detectors directed in the same way (horizontally or vertically) and installed in the same height?*

AR: *The same horizontal detector arrangement was used in both experiments. The height above the surface is about 20 cm lower for the hand-wagon compared to the car. We will add some details to the methods section.*

**1.33. Page 15 L20-23**

RC: *Could the model results be scaled to match the CRNS roving measurements?*

AR: *In principle yes, but comparative quantification of the model and observation results is beyond the focus of this study (as explained above) and would require more refined model design.*

**1.34. Figure 8**

RC: *What is the outline of the measured area in Figures 8D and 8E representing? Is it the driven route (I guess not, since the line crosses through building) or is it some meters away from where you drove/walked the rover (e.g., the footprint radius)? Furthermore, what is the uncertainty of the measurements and modeling? Please provide these number in the figure or in the text.*

AR: *The area around the measurement points represents the envoloping geometry with distances of 30–50m from the path. We will add information to measurement uncertainty to the revised manuscript.*

**1.35. Page 17 L26–Page 18 L2**

RC: *Please revise this paragraph. Interception, as an important variable for urban hydrology, should be mentioned when explaining urban hydrology in the introduction. The authors conclude that the difference between CRNS and WSN soil moisture is due to interception. Evidence is missing that the additional water seen by the CRNS detector is interception, and the difference could just as well be a result of other*

*effects, e.g., different measurement scale (point scale and hectometer scale) and measurement depth. Either add some supporting modeling and measurement results or exclude the part of interception from the manuscript. A third option could be to include "interception" in the discussion of the results provided in Figure 9b. The discussion should also include other possible reasons for the different soil moisture of CRNS and WSN.*

AR:   *Thank you for these suggestions. We will rewrite the paragraph to emphasize also other possible explanations for the excess water storage seen by the detector during and after rain events. However, we think that intercepted water (either over sealed or unsealed surfaces) is the most likely explanation, and we disagree with the reviewer concerning the other options. The different horizontal scale of the measurement is an unlikely explanation, as this would require other parts of the planar footprint area to accumulate much more soil water than the grassland meadow. We consider also the effect of the measurement depth as an unlikely explanation, as those effects have been taken into account upon averaging the WSN data (weighting of the six WSN sensors from 0 to 40 cm according to neutron transport theory).*

**1.36.    Page 18 L3-9**

RC:   **Either, explain the purpose of the field experiment with greater detail as well as show and discuss the results, or take this part out.**

AR:   *The purpose of the experiment was to quantify the effect of nearby irrigation to the neutron sensors. We were able to predict the results with the theoretical calculation presented here. In the revision we will add further details and consider adding a figure of the measurement results.*

---

## Referee Comment (RC2) · Anonymous Referee #1 · 20 Sep 2017

The authors had addressed my concerns very well. A nice technical contribution to advancing CRNP method.
* * *